

# Genetic divergences and hybridization within the *Sebastes inermis* complex

Diego Deville[1], Kentaro Kawai[1], Hiroki Fujita[2] and Tetsuya Umino[1]

[1] Graduate School of Integrated Sciences for Life, Hiroshima University, Higashihiroshima, Hiroshima, Japón
[2] Seto Marine Biological Laboratory, Field Science Education and Research Center, Kyoto University, Shirahama, Wakayama, Japan

Corresponding author
Tetsuya Umino,
umino@hiroshima-u.ac.jp

## ABSTRACT

The *Sebastes inermis* complex includes three sympatric species (*Sebastes cheni*, viz *Sebastes inermis*, and *Sebastes ventricosus*) with clear ecomorphological differences, albeit incomplete reproductive isolation. The presence of putative morphological hybrids (PMH) with plausibly higher fitness than the parent species indicates the need to confirm whether hybridization occurs within the complex. In this sense, we assessed the dynamics of genetic divergence and hybridization within the species complex using a panel of 10 microsatellite loci, and sequences of the mitochondrial control region (D-loop) and the intron-free rhodopsin (RH1) gene. The analyses revealed the presence of three distinct genetic clusters, large genetic distances using D-loop sequences, and distinctive mutations within the RH1 gene. These results are consistent with the descriptions of the three species. Two microsatellite loci had signatures of divergent selection, indicating that they are linked to genomic regions that are crucial for speciation. Furthermore, nonsynonymous mutations within the RH1 gene detected in *S. cheni* and "Kumano" (a PMH) suggest dissimilar adaptations related to visual perception in dim-light environments. The presence of individuals with admixed ancestry between two species confirmed hybridization. The presence of nonsynonymous mutations within the RH1 gene and the admixed ancestry of the "Kumano" morphotype highlight the potential role of hybridization in generating novelties within the species complex. We discuss possible outcomes of hybridization within the species complex, considering hybrid fitness and assortative mating. Overall, our findings indicate that the genetic divergence of each species is maintained in the presence of hybridization, as expected in a scenario of speciation-with-gene-flow.

## INTRODUCTION

Speciation-with-gene-flow is postulated to occur in a scenario involving divergent selection, in which populations adapt to diverse environments and reach distinct fitness optima. In this scenario, prezygotic and extrinsic postzygotic isolating barriers are expected to appear first, working together to reduce hybridization between species, while

intrinsic postzygotic isolating barriers might appear later, enforcing the reproductive isolation of species (*Seehausen et al., 2014*). If hybridization occurs, the possible outcomes in speciation primarily depend on the fitness of hybrids related to the parent species and their reproductive success in specific environments (*Baskett & Gomulkiewicz, 2011*; *Servedio & Hermisson, 2020*). Accordingly, a higher fitness of hybrids may have a significant evolutionary potential to generate novel lineages and/or adaptations (*Arnold & Fogarty, 2009*; *Abbott et al., 2013*), leading to the maintenance of incomplete reproductive isolation under certain circumstances of assortative mating (*Servedio & Hermisson, 2020*). Conversely, if hybrids have lower fitness than the parent species, hybridization can contribute to increasing reproductive isolation of the hybridizing lineages (*i.e.*, reinforcement) (*Bank, Hermisson & Kirkpatrick, 2012*). Thus, the occurrence of hybridization in a single clade offers the possibility of directly assessing hybrid fitness and the significant contribution of its possible outcomes to speciation.

*Sebastes* is a genus whose diversification is driven mainly by divergent selection (*Ingram, 2011*). Several cases of hybridization within this genus have been inferred using various methods. For example, morphological analyses have been used to identify hybrids endowed with morphotypes that are intermediate to their parent species (*Valentin, Sévigny & Chanut, 2002*; *Muto et al., 2013*). Hybridization events have also been detected using population genetic surveys and Bayesian clustering methods (*Roques, Sévigny & Bernatchez, 2001*; *Buonaccorsi et al., 2005*; *Burford, 2009*; *Saha et al., 2017*; *Keller et al., 2022*). These methods are widely used because they can infer the number of distinct genetic clusters and estimate the admixture proportions of individuals in them (*Pritchard, Stephens & Donnelly, 2000*; *Porras-Hurtado et al., 2013*; *Thia, 2023*).

The species complex *Sebastes inermis* encompasses three species: viz. *Sebastes inermis* Cuvier, 1829 (red rockfish), *Sebastes cheni* Barsukov, 1988 (brown to golden-brown rockfish, known as "white" in Japan), and *Sebastes ventricosus* Temminck & Schlegel, 1843 (greenish to black rockfish). They are slow-moving species with sympatric occurrence along the coastal waters of Japan, particularly in rocky reefs and beds of *Zostera* L. and *Sargassum* C. Agardh, 1820 (*Kai & Nakabo, 2008*). Although clear stock limits for these species have not been defined, their economic significance for local communities has prompted the annual release of thousands of juveniles to enhance populations (*Nakagawa, 2008*). In addition to their different body colours, the morphological identification of these rockfishes mainly relies on meristic counts and body proportions (*Kai & Nakabo, 2008*), and differences in otolith descriptors and body shape can facilitate their identification (*Deville et al., 2023*). These rockfishes have different growth rates, with *S. cheni* attaining larger body sizes than *S. ventricosus*, which is larger than *S. inermis* at the same age (*Kamimura et al., 2014*). The morphological divergences of these species suggest asymmetric depth distributions, which can reduce their interspecific competition and allow their coexistence in sympatry (*Deville et al., 2023*). Genetic identification of the three rockfishes can be accomplished by examining allele differences in amplified fragment length polymorphisms (AFLP) (*Kai, Nakayama & Nakabo, 2002*; *Kai & Nakabo, 2008*) and two microsatellite loci (*Deville et al., 2023*). Dissimilar alleles in these molecular markers suggest reproductive isolation of these species (*Kai, Nakayama & Nakabo, 2002*).

Prezygotic reproductive barriers, such as differences in acoustic and visual communication systems, are expected to sustain the reproductive isolation of these species (*Deville et al., 2023*). The three species are single brooders, and their reproductive seasons occur during the winter months (*Plaza, Katayama & Omori, 2004*). Polygamous individuals have been observed and inferred using paternity tests (*Shinomiya & Ezaki, 1991*; *Blanco Gonzalez et al., 2009*).

Putative morphological hybrids (PMH) displaying intermediate colourations and meristic counts but with otolith descriptors of *S. cheni* have been reported in the Seto Inland Sea (Hiroshima Prefecture) (*Deville et al., 2023*). Additionally, an endemic intermediate morphotype of *S. cheni* and *S. inermis* (colloquially called "big red") has been reported by local fishermen in Kumano Nada (Wakayama Prefecture) but without any genetic information. Mating behaviour suggests that these PMH might have a higher fitness than *S. inermis* and *S. ventricosus* during reproductive seasons because (1) in contrast to smaller males, larger males perform agonistic behaviour and courtship rituals, and occupy wider territories, and (2) females tend to mate with larger males (*Shinomiya & Ezaki, 1991*). Therefore, there is an urgent need to confirm whether hybridization occurs within the *S. inermis* complex. By confirming hybridization, it would also be possible to determine the impact of this process on the population structure of each species because hybridization can alter estimates of genetic diversity (*Berntson & Moran, 2009*; *Artamonova et al., 2013*; *Saha et al., 2017*).

To investigate the dynamics of genetic divergence and hybridization within the *S. inermis* complex, we employed genetic information from 10 microsatellite loci, and sequences of both the mitochondrial control region (D-loop) and the intron-free rhodopsin (RH1) gene. Microsatellite loci are highly polymorphic nuclear markers that have been used to discriminate closely related species within *Sebastes* (*e.g.*, *Frable et al., 2015*; *Saha et al., 2017*; *Bizzaro et al., 2020*; *Keller et al., 2022*). By utilizing these markers, it is possible to assess whether any locus is under putative divergent selection. This is achieved by detecting anomalously high interspecific differences in contrast to low intraspecific differences (called the "FST outlier" approach) (*Beaumont & Balding, 2004*). D-loop was used since this region offers enough resolution to obtain significant genetic distances among the three species (*Kai & Nakabo, 2008*). The RH1 gene was analysed to identify any genetic basis for the suggested asymmetric depth distributions of the three species (*Deville et al., 2023*), given that nonsynonymous mutations in this gene can indicate that species inhabit environments with varying levels of downwelling sunlight along the water column (*Sivasundar & Palumbi, 2010*; *Shum et al., 2014*).

Our study aimed to address the following objectives:

1) Assess the genetic divergences between species using sympatric individuals with clear morphological distinction. This will allow us to increase the evidence supporting the genetic divergence of the three species.

2) Investigate whether any of the microsatellite loci have signatures of divergent selection. When a locus is under divergent selection, alleles of any locus near linked regions will also be under divergent selection; thus, that selection will prevent gene flow in all nearby

**Table 1 Number of individuals of each species and putative morphological hybrid collected in each sampling location.**

| Morphotype | Akita | Hiroshima | Kagoshima | Wakayama | Total |
|---|---|---|---|---|---|
| *S. ventricosus* | | 41 | 28 | 33 | 102 |
| *S. inermis* | | 42 | 32 | 37 | 111 |
| *S. cheni* | 30 | 43 | 3 | 13 | 89 |
| Black-white | | 22 | 1 | 3 | 26 |
| Red-white | | 19 | 7 | | 26 |
| Kumano | | | | 6 | 6 |
| Total | 30 | 167 | 71 | 94 | 360 |

genomic regions, leading to a reduction in the migration rate (*i.e.*, gene flow) of that region (*Feder, Egan & Nosil, 2012*).

3) Infer the presence of hybrids by using clustering analyses of microsatellite loci. This can confirm whether the PMH are genetically hybrid individuals.

4) Evaluate the level of genetic divergence of the PMH.

We hypothesized that each species maintains its genetic divergence even in the presence of hybridization, as expected in a scenario of speciation-with-gene-flow (*Feder, Egan & Nosil, 2012*). We also anticipated that the PMH would exhibit genetic signatures consistent with the admixture of the parent species. We discuss how assortative mating and the relative fitness of hybrids interact to maintain the divergence of species in the presence of hybridization.

## MATERIALS AND METHODS

### Sampling, DNA isolation, and sequencing

We examined 360 individuals collected *via* bank fishing on rocky and sandy shores along the coast of Japan (Table 1 and Fig. 1). Species identification was carried out following the criteria established by *Kai & Nakabo (2008)*, starting from colour alive and fresh, meristic counts, and body proportions. In addition, the otolith weight~age relationship was used to improve the identification of specimens older than 3 years, as suggested by *Deville et al. (2023)*.

Individuals were categorized into six different morphological groups, depending on whether they had all the diagnostic traits of any species without overlap, or only some of them (Table 1 and Fig. 2). The categories were as follows (Fig. 2): (1) white *S. cheni*, (2) red *S. inermis*, (3) black *S. ventricosus*, (4) PMH black-white (BW) *S. cheni* x *S. ventricosus*, with some individuals exhibiting colouration from one species while their meristic counts and body proportions resemble the ones of the other species, (5) PMH red-white (RW) *S. cheni* x *S. inermis*, with two slightly different intermediate colourations, intermediate meristic counts but otolith weight~age relationships of *S. cheni*, and (6) "Kumano" or "big red" morphotype collected in sandy and rocky shores in East Wakayama Prefecture. It is considered a hypothetical hybrid of *S. inermis* and *S. cheni* because it displays intermediate

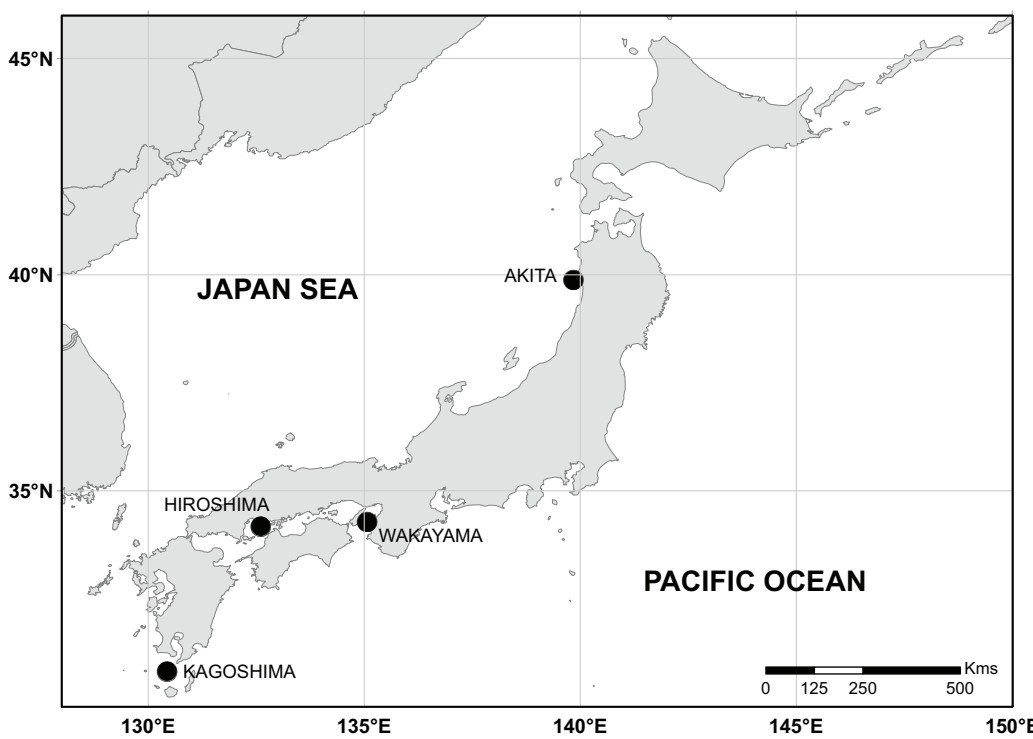

**Figure 1 Sampling sites along coastal waters of Japan.** Black points represent sampling sites. The reference map was retrieved from the Natural Earth website (https://www.naturalearthdata.com/). Data contained in this website is in the public domain.

colouration and meristic counts of the two species but an otolith weight~age relationship resembling that of *S. cheni*. Additional photos of reference specimens for each species and PMH are shown in Fig. S1.

Total DNA was isolated from a small piece of pectoral fin using the TNES-urea buffer (*Asahida et al., 1996*) followed by the standard phenol-chloroform isolation. A set of 10 microsatellite loci isolated from *Sebastes schlegelii* Hilgendorf, 1880 (SSC12, SSC23, KSs2A, KSs6, KSs7, and CGN1) (*Yoshida, Nakagawa & Wada, 2005*; *An et al., 2009*; *Gao et al., 2018*), *S. inermis* (Sebi1, Sebi2, and Sebi3) (*Blanco Gonzalez et al., 2009*), and *Sebastes rastregiller* (Jordan & Gilbert, 1880) (SRA7-7) (*Westerman et al., 2005*) were cross-amplified by multiplex PCRs. The four universal primers proposed by *Blacket et al. (2012)* were labelled with 6-FAM (Tail A), VIC (B), NED (C), and PET (D), while the forward primers of all loci were modified at their 5′ ends with the same universal primers (Table S1). Standardization of multiplex PCRs was performed as described by *Deville et al. (2021)*. Each multiplex PCR was carried out in a volume of 5 μL containing 2.5 μL of 2× KOD Fx Neo buffer, 1 μL of dNTP 2 μM, 0.1 μL of 1U KOD polymerase (Toyobo Co., Ltd., Osaka, Japan), 1 μL of DNA 50 ng/μL, 0.3 μL of ddH$_2$O, and 0.1 μL of a primer mix (5 mM labelled universal primers and modified forward primers, and 10 mM reverse primers). Multiplex PCRs were performed in a Mastercycler Gradient 96-well system (Eppendorf, Hamburg, Germany) with initial denaturation at 94 °C for 4 min followed by a touchdown (10 cycles at 94 °C/1 min, annealing from 63 °C to 54 °C/1 min and 72 °C/1 min), 20 cycles

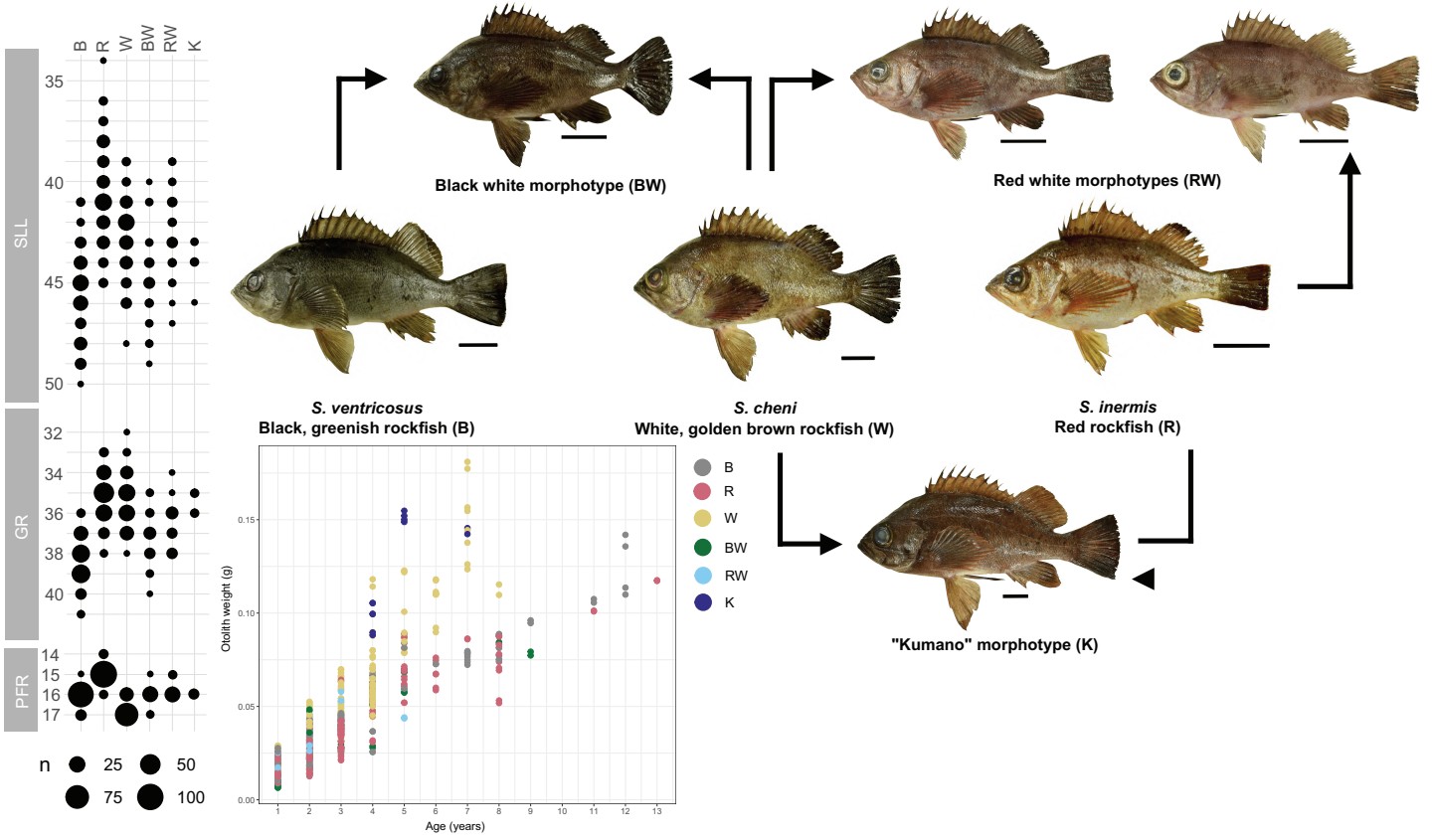

**Figure 2 Colouration patterns, meristic counts, and otolith weight~age relationships of the three rockfishes** *Sebastes cheni, Sebastes inermis,* **and** *Sebastes ventricosus,* **and the putative morphological hybrids between them.** The frequency distributions of the number of pored lateral line scales (SLL), number of gill rakers of the first arch (GR), and number of radials of the pectoral fin (PFR) are indicated in each species and putative morphological hybrid. Reference sizes for frequencies are indicated below the three variables. Points in the otolith weight~age plot were coloured to ease distinction of species and putative morphological hybrids. Arrows connecting specimens indicate the putative origin of each morphological hybrid. A scale of 3 cm was added next to each specimen as reference for size.

with the same conditions but annealing at 55 °C, and a final step of 72 °C during 10 mins. Then, 1 μL of the PCR product was mixed with 18.8 μL of Hi-DiTM Formamide (Applied Biosystem, Waltha, MA, USA) and 0.2 μL of GeneScanTM-600 LIZ® size standard (Applied Biosystem, Waltha, MA, USA). This mixture was denatured at 95 °C for 3 min and run on an ABI 3,130×1 Genetic Analyzer (Applied Biosystem, Waltha, MA, USA). Genotyping was performed using GeneMarker v.2.6® (Soft Genetics, State College, PA, USA).

D-loop and RH1 gene were amplified in 124 and six individuals from Hiroshima and Wakayama, respectively. This includes (1) 25 individuals of each species identified through morphological and genetic analyses (clustering analysis, see Fig. 3), (2) eight individuals morphologically assigned to a species but genetically classified as putative hybrids, (3) 22 individuals with the BW morphotype, (4) 19 specimens showing the RW morphotype, and (5) six "Kumano" specimens from Wakayama. D-loop was amplified using the MebTD1F forward (5′→3′: ACCTGAATCGGAGGAATGCC) and MebTD1R reverse (5′→3′: GGGTTTACAGGAGCGTTAGC) primers. These primers were designed using the

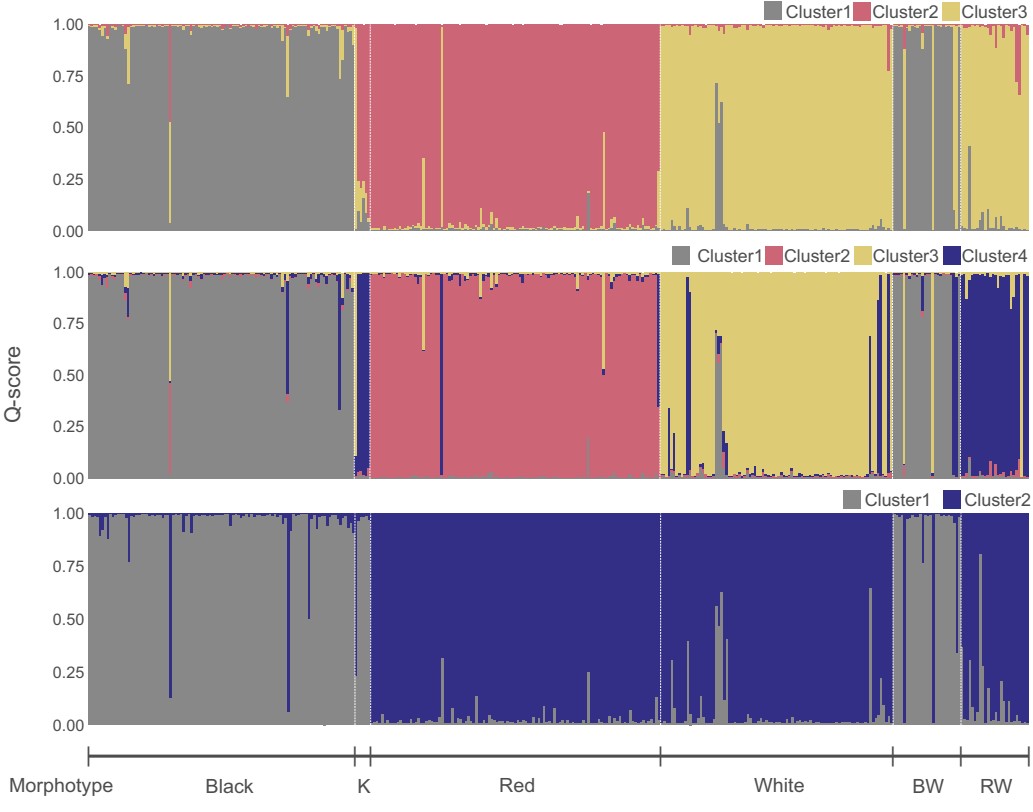

**Figure 3 Genetic clusters inferred in the *Sebastes inermis* complex using ten (above and middle) and eight microsatellite loci (below).** Individuals are coloured based on their ancestry coefficients (Q-score) for each genetic cluster. Putative morphological hybrids are indicated as K ("Kumano"), BW (*S. cheni* × *S. ventricosus* morphotype), and RW (*S. cheni* × *S. inermis* morphotype).

mitochondrial genomes of *S. inermis* (NC_023456), *S. schlegelii* (NC_005450), and *Sebastes thompsoni* (Jordan & Hubbs, 1925) (KJ834064) as references. The RH1 gene was amplified using the Rh193 (5′→3′: CNTATGAATAYCCTCAGTACTACC) and Rh1039r (5′→3′: TGCTTGTTCATGCAGATGTAGA) primers (*Chen, Bonillo & Lecointre, 2003*). Both genetic regions were amplified in a total volume of 8 μL containing 4 μL of 2× KOD buffer, 1.2 μL of dNTP 2 μM, 0.1 μL of each primer at 10 mM, 0.1 μL of 1U KOD Taq polymerase, 1 μL of DNA 50 ng/μL, and 1.5 μL of ddH$_2$O. PCR conditions for both genetic regions were as follows: initial denaturation at 94 °C for 4 mins; 35 cycles of 94 °C for 20 s, 55 °C for 30 s, and 68 °C for 45 s; and a final extension at 68 °C for 5 mins. Each PCR product was cleaned up using ExoSAP-IT (Affymetrix/USB Corporation, Cleveland, OH, USA) and then sequenced using the BigDye v3.1 Terminator Sequencing Kit (Applied Biosystems, Waltham, MA, USA) on a Genetic Analyzer ABI 3,130×1 (Applied Biosystems, Waltham, MA, USA). D-loop amplicons were sequenced using the MebTD1F primer, whereas RH1 amplicons were sequenced in both directions whenever an ambiguous nucleotide was found in the chromatograms. Chromatograms were visualized and manually edited using Chromas Lite v2.6.6 (Technelysium Pty. Ltd., Brisbane, QLD, Australia), and the sequences were aligned using Clustal X2 (*Larkin et al., 2007*). RH1

sequences were phased into two sequences per individual using the program PHASE implemented in DNAsp v6 (*Rozas et al., 2017*) with a Markov chain Monte Carlo of 100,000 iterations, burn-in of 10,000 steps, and 10-step thinning intervals.

## Descriptive statistics and genetic divergences

Descriptive statistics were estimated for only eight populations containing more than 25 individuals (two populations of *S. cheni* and three populations of each *S. inermis* and *S. ventricosus*) (Table 1). The occurrence and percentages of null alleles were evaluated using Micro-Checker v.2.2.3 (*Van Oosterhout et al., 2004*) and the formula of *Brookfield (1996)*, respectively. We estimated the number of alleles ($N_A$), observed heterozygosity ($H_O$), and expected heterozygosity ($H_E$) for each population. Exact tests for Hardy-Weinberg equilibrium (HWE) and linkage disequilibrium were performed for each locus and pair of loci, respectively. Pairwise genetic distances between species (FST) were estimated based on the number of different alleles (*Weir & Cockerham, 1984*). The FST values were estimated within (1) the eight previously mentioned populations, and (2) the genetically distinct individuals and PMH in which D-loop and the RH1 gene were sequenced. All analyses were performed using Arlequin v3.5 (*Excoffier & Lischer, 2010*). Because null alleles can inflate FST estimates, we used FreeNA (*Chapuis & Estoup, 2007*) to calculate unbiased FST after null allele correction with the ENA method. Confidence intervals of the corrected FST were obtained using a bootstrap resampling procedure performed 1,000 times.

The D-loop and phased RH1 sequences were collapsed into haplotypes, and the number of haplotypes, haplotype diversity, and nucleotide diversity were estimated using DNAsp v6. The frequencies of D-loop haplotypes were used to estimate FST values between species and PMH using the K2P model (K2P distances) (*Kimura, 1980*) in Arlequin v3.5. Networks of D-loop and RH1 haplotypes were constructed using the TCS method (*Clement, Posada & Crandall, 2000*) implemented in PopArt v1.7 (*Leigh & Bryant, 2015*). To determine the position of mutations occurring in the RH1 sequences and their possible relationships with changes in the protein function, we selected the individual with the longest sequence in each species and pooled them together with the publicly available RH1 sequences of 36 *Sebastes* rockfishes (EF212407–EF212438, KM013899, KM013904, KM013924, and KM013927). For this analysis, the complete amino acid sequence of the bovine RH1 (NM_001014890) was used as a reference.

## Detection of outlier loci

Outlier loci with very low or high divergence among species were detected using BayeScan v2.1. (*Foll & Gaggiotti, 2008*). The analysis was performed only with allele frequencies of individuals assigned to a species based on their morphology. The parameters for the analysis were as follows: 100,000 burn-in steps, a thinning interval of 100, a sample size of 10,000, 50 pilot runs, a pilot length of 10,000, and a value of 10 for prior odds. The analysis assesses selection using logistic regression. It decomposes FST coefficients into a population-specific component ($\beta$) shared by all loci, and a locus-specific component ($\alpha$) shared by all populations. Loci under selection are inferred when FST coefficients are

largely explained by the locus-specific component (*i.e.*, α is significantly different from 0). Positive α values indicate divergent selection, whereas negative values suggest balancing or purifying selection. The significance of each α value per locus was evaluated by checking corrected *P* values calculated using the False Discovery Rate method (FDR) (*Benjamini & Hochberg, 1995*).

## Genetic clusters and individual admixture analysis from microsatellite loci

Genetic clustering was assessed using STRUCTURE v2.3.4 (*Pritchard, Stephens & Donnelly, 2000*). The analysis estimates the number of homogeneous genetic clusters (K) that maximize Hardy-Weinberg and linkage equilibrium (*Pritchard, Stephens & Donnelly, 2000*), and then calculates individual admixture proportions (Q-score = genome ancestry coefficient). The inference of Q-scores performed in STRUCTURE is facilitated by using markers showing different alleles at very high frequencies in distinct populations (*Porras-Hurtado et al., 2013*), which can be expected in loci under putative divergence. Although the inclusion of these loci can be problematic, combining them with other loci helps to balance out selection, as selection is not expected to fix alleles across independent loci scored in different populations (*Selkoe & Toonen, 2006*). Hence, two analyses were performed: the first using all microsatellite loci, and the second excluding loci under putative divergent selection. STRUCTURE analyses were run with a Markov chain Monte Carlo of 1,000,000 steps, 10% burn-in, an independent allele frequency model, K values from 1 to 7, and 10 replicates for each K value. The most likely number of genetic clusters was inferred using the Evanno method (*Evanno, Regnaut & Goudet, 2005*), as implemented in STRUCTURE HARVESTER (*Earl & vonHoldt, 2012*). To infer the number of genetic clusters without the assumptions of HWE and linkage equilibrium (assumed in STRUCTURE), we performed a Discriminant Analysis of Principal Component (DAPC) implemented in the Adegenet 2.1.1 package (*Jombart, Devillard & Balloux, 2010*). The optimal number of genetic clusters was determined by considering the distribution of individuals projected into the space of principal components and the "elbow point" in the distribution of the Bayesian information criterion (BIC) scores (*Thia, 2023*). For DAPC, we only preserved "K-1" principal components because they capture the maximal among-species variation without adding an unplanned interpretation of the population structure (*Thia, 2023*). It is possible to assign individuals to the inferred genetic clusters or categorize them as putative hybrids, considering a threshold Q-score. Here, individuals were categorized as putative genetic hybrids if they had Q-scores lower than 0.9, following *Sanz et al. (2009)*.

For each species, a "reference population" of 30 individuals with clear morphological distinction and Q-scores higher than 0.99 was established. These populations were used to simulate putative pure individuals, putative first-generation (F1) hybrids, and putative backcrosses in HYBRIDLAB v1.0 (*Nielsen, Bach & Kotlicki, 2006*). A total of 810 putative pure parental genotypes were generated by simple mechanical mixing of the alleles from each "reference population". For putative F1 hybrids, we simulated 30 individuals from each parent cross: *S. cheni* × *S. inermis*, *S. cheni* × *S. ventricosus*, and *S. inermis* ×

**Table 2 Descriptive statistics for each microsatellite locus in the three species.**

| Locus | S. cheni (N = 86) | | | S. inermis (N = 111) | | | S. ventricosus (N = 102) | | |
|---|---|---|---|---|---|---|---|---|---|
| | Na | Ho | He | Na | Ho | He | Na | Ho | He |
| SSC12 | 7 | 0.539 | 0.531 | 8 | 0.703 | 0.743 | 6 | 0.725 | 0.726 |
| Sebi1 | 8 | 0.730 | 0.672 | 26 | 0.631 | 0.654 | 70 | 0.951 | 0.971 |
| KSs2A | 17 | 0.719 | 0.746 | 41 | 0.730 | 0.931 | 22 | 0.775 | 0.903 |
| Sebi3 | 18 | 0.888 | 0.893 | 11 | 0.838 | 0.855 | 15 | 0.912 | 0.889 |
| SSC23 | 8 | 0.629 | 0.605 | 14 | 0.838 | 0.798 | 9 | 0.696 | 0.727 |
| **KSs7** | 6 | 0.494 | 0.581 | 10 | 0.622 | 0.722 | 7 | 0.657 | 0.485 |
| Sebi2 | 6 | 0.348 | 0.366 | 6 | 0.559 | 0.551 | 5 | 0.500 | 0.558 |
| SRA7-7 | 16 | 0.876 | 0.873 | 15 | 0.793 | 0.830 | 15 | 0.804 | 0.838 |
| KSs6 | 21 | 0.831 | 0.895 | 14 | 0.892 | 0.898 | 15 | 0.765 | 0.848 |
| **CGN1** | 11 | 0.584 | 0.752 | 8 | 0.649 | 0.679 | 6 | 0.480 | 0.594 |
| Mean | 11.8 | 0.664 | 0.691 | 15.3 | 0.725 | 0.766 | 17 | 0.726 | 0.754 |

**Note:**

N, sample size; $N_A$, number of alleles; $H_O$, observed heterozygosity; $H_E$, expected heterozygosity. Loci in bold font are under putative divergent selection.

*S. ventricosus*. For the putative backcrosses, 30 individuals were simulated for each putative pure parental and putative F1 cross, resulting in six groups of backcrosses. Each group was labelled with three letters depending on the colouration of each parent species (*i.e.*, black: B, red: R, and white: W) to ease their distinction. The first and second letters indicate the putative F1 hybrid origin of the first parent, and the third letter indicates the putative pure origin of the second parent. The six groups of backcrosses were BRB, BRR, BWB, BWW, RWR, and RWW. The numbers of simulated putative pure individuals, putative F1 hybrids, and putative backcrosses were set to ensure 10% of putative hybrids in our simulated dataset to effectively infer hybrids using clustering analyses (*Vähä & Primmer, 2006*). We pooled the simulated individuals in a single dataset and estimated the number of genetic clusters and Q-scores using STRUCTURE with the same parameters as those used in the analyses with our collected samples. The maximum Q-scores of all individuals included in the observed and simulated datasets were plotted to detect putative F1 hybrids and backcrosses.

# RESULTS

## Descriptive statistics from microsatellite loci

Micro-Checker indicated that the loci KSs2A and CGN1 presented null alleles in more than two populations, which was corroborated by bootstrapped estimations of the percentages of null alleles (Fig. S2). Individuals presenting homozygous alleles were re-amplified to confirm allele sizes. All loci were polymorphic within each species, with the number of alleles per locus ranging from six to 21 in *S. cheni*, six to 41 in *S. inermis*, and five to 70 in *S. ventricosus* (Table 2). The mean $H_O$ values of *S. cheni*, *S. inermis*, and *S. ventricosus* were 0.664, 0.725, and 0.726, respectively. The mean value of $H_E$ was 0.691 for *S. cheni*, 0.766 for *S. inermis*, and 0.754 for *S. ventricosus*. Significant differences in allele

**Table 3 Genetic distances (FST) for populations of the three species estimated from allele frequencies of 10 microsatellite loci.**

| Population | AW | HB | HR | HW | KB | KR | WB | WR |
|---|---|---|---|---|---|---|---|---|
| AW | | 0.164 | 0.113 | 0.009 | 0.149 | 0.122 | 0.174 | 0.127 |
| HB | 0.167 | | 0.137 | 0.124 | 0.036 | 0.142 | 0.013 | 0.139 |
| HR | 0.116 | 0.143 | | 0.100 | 0.122 | 0.017 | 0.133 | −0.002 |
| HW | 0.010 | 0.127 | 0.105 | | 0.111 | 0.109 | 0.133 | 0.114 |
| KB | 0.152 | 0.035 | 0.125 | 0.111 | | 0.130 | 0.018 | 0.129 |
| KR | 0.125 | 0.148 | 0.018 | 0.114 | 0.133 | | 0.141 | 0.016 |
| WB | 0.176 | 0.013 | 0.137 | 0.134 | 0.016 | 0.145 | | 0.135 |
| WR | 0.128 | 0.144 | −0.002 | 0.117 | 0.131 | 0.018 | 0.138 | |

Note:
FST values without correction of null alleles are indicated below the diagonal. FST values with corrected null allele frequencies using the ENA method in FreeNA are shown above the diagonal. The first and second letter of the abbreviations of populations indicate the sampling place (A, Akita; H, Hiroshima; K, Kagoshima; W, Wakayama) and the three species (B, black rockfish; R, red rockfish; W, white rockfish), respectively.

distributions (Fig. S3) were detected in nine of 10 microsatellite loci ($P$ values < 0.005), with 23 of 30 pairwise comparisons being statistically significant ($P$ values < 0.002). Significant deviations from HWE expectations were detected in six tests, three of which occurred at locus CGN1 ($P$ values < 0.005) (Table S2). Only one of the 360 pairwise comparisons of loci (SSC23-Sebi2 in *S. inermis* from Hiroshima) showed significant linkage disequilibrium ($P$ value < 0.001). Significant FST values higher than 0.1 were found between species in both the uncorrected and corrected null allele frequencies (Table 3). This indicates a meaningless effect of null allele frequencies on the estimates of genetic divergence.

## Outlier microsatellite loci

Two loci (KSs7 and CGN1) were under putative divergent selection ($\alpha$-KSs7 = 1.38, $\alpha$-CGN1 = 1.50, adjusted $P$ values using FDR < 0.002). In addition, the locus Sebi2 presented some signatures of putative divergent selection ($\alpha$ = 0.89), but this was not statistically significant (adjusted FDR $P$ value > 0.08). The FST values among species calculated from CGN1, KSs7, and Sebi2 were 0.233, 0.215, and 0.15, respectively. The FST values from these loci were much higher than the mean value of 0.06 in the other seven microsatellite loci (Table S3).

## Genetic clusters and individual admixture analysis

The most likely number of genetic clusters was three (Figs. 3 and 4), as suggested by the Evanno method ($\Delta k$ = 795.07) (Fig. S4), the distribution of individuals projected into the space of principal components (Fig. 4B), and the distribution of BIC scores (Fig. 4C). This indicates that deviations from HWE and linkage equilibrium of some loci did not influence the inference of genetic clusters in this study. The results of STRUCTURE inferring four genetic clusters, separated the three species, and lumped RW and "Kumano" individuals into a single cluster (Fig. 3). Meanwhile, clustering analyses, excluding the two loci under

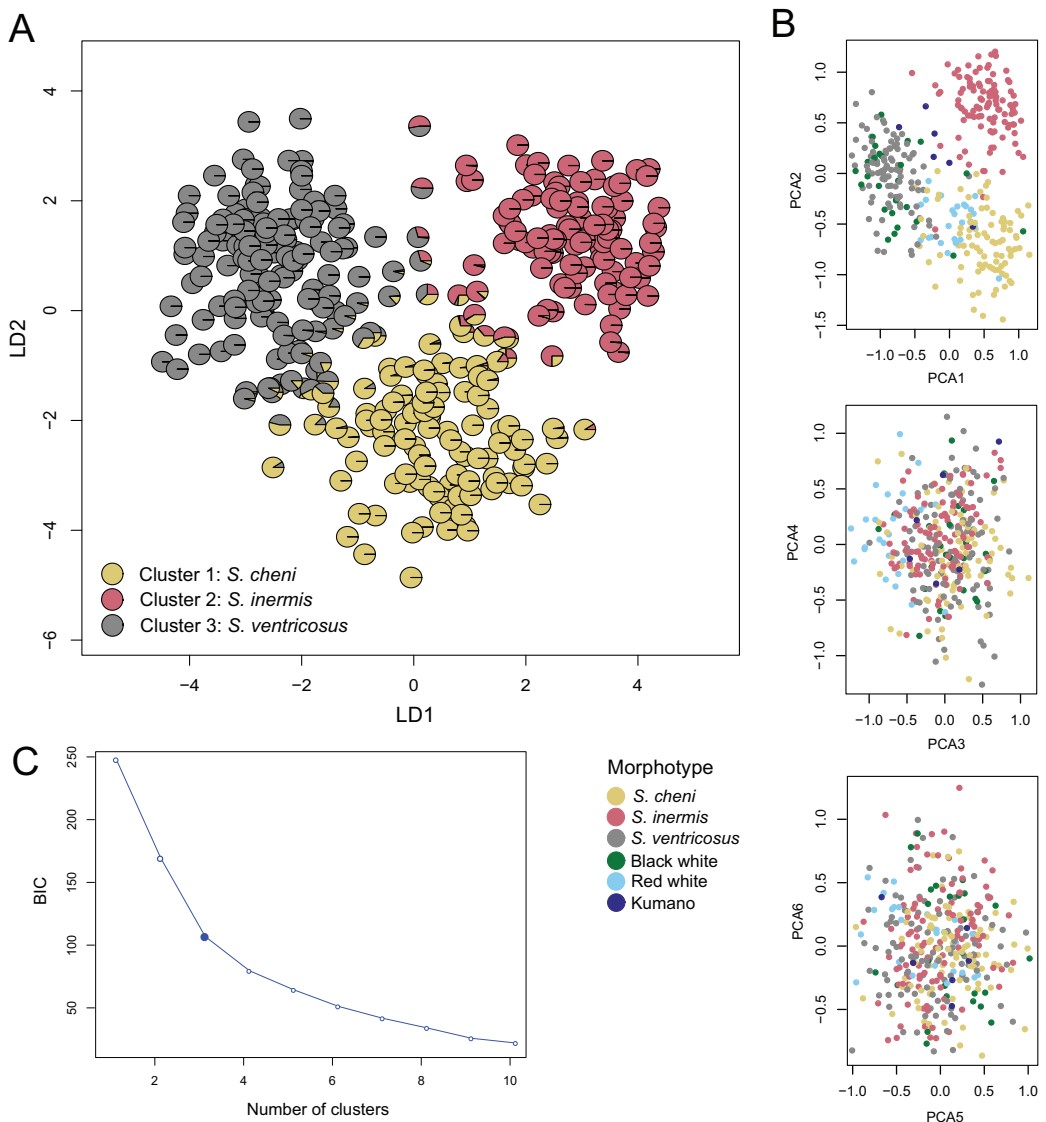

**Figure 4 Discriminant analysis of principal components (DAPC).** (A) Scatterplot of individuals in the discriminant functions (LD1 and LD2). Individuals are represented by pie charts showing their posterior probabilities. Colours of the genetic clusters are indicated in the legend. (B) Scatterplots of individuals projected into the space of principal components (PCs). Individuals were coloured based on their morphological features. (C) Inferred number of genetic clusters by k-means clustering considering the "elbow" point in the distribution of the Bayesian information criterion (BIC) scores.

significant divergent selection, identified two clusters (Δk = 1224.42) (Fig. S5), which only allowed discrimination of *S. ventricosus* from the other two species (Fig. 3).

A total of 331 individuals were assigned as genetically distinct (Q-score > 0.90), and 29 as putative genetic hybrids. Among the BW individuals, two, three, and 21 were categorized as putative hybrids, *S. cheni*, and *S. ventricosus*, respectively. The RW group contained 20 and six individuals that genetically qualified as *S. cheni* and putative hybrids, respectively. Among the six "Kumano" individuals, three had admixed ancestry of *S. cheni*

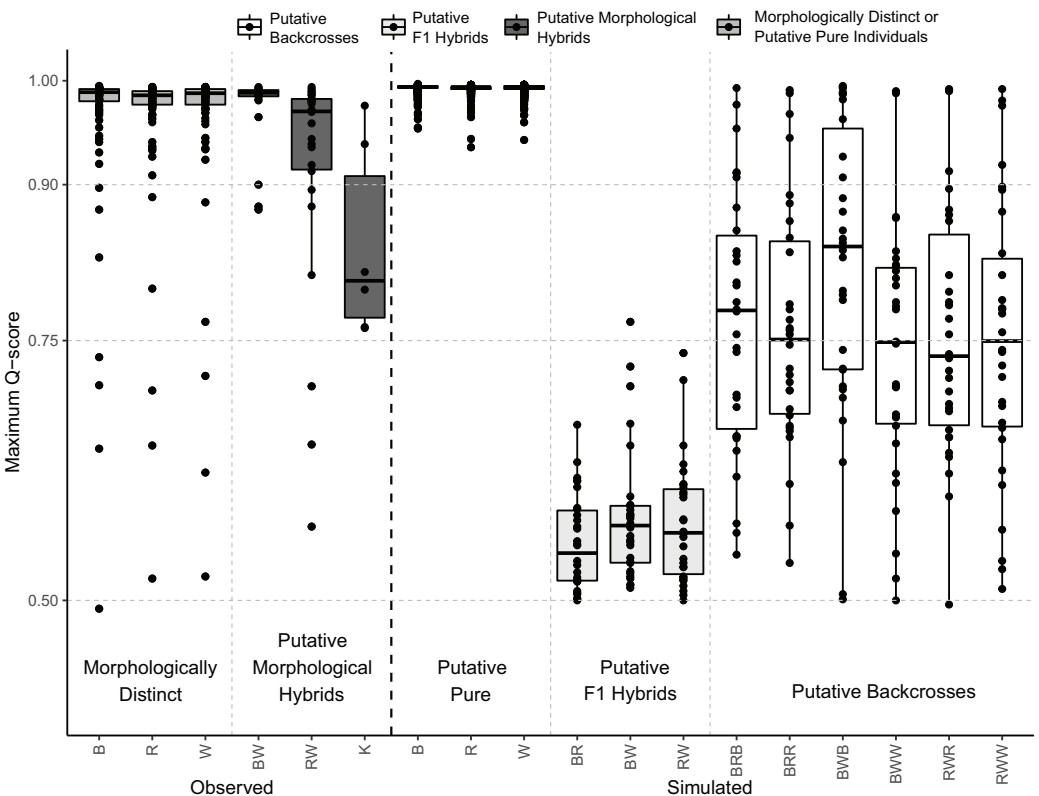

**Figure 5 Distribution of maximum Q-scores calculated in STRUCTURE using the observed and simulated individuals.** Central bold lines in the box plots indicate the medians; box limits represent the 1st and 3rd quartiles; Q-scores are drawn as black circles. Different colours indicate whether boxplots are from morphologically distinct or putative pure individuals, putative morphological hybrids, putative F1 hybrids, or putative backcrosses as represented in the legend above. B: black rockfish (*S. ventricosus*), R: red rockfish (*S. inermis*), W: white rockfish (*S. cheni*), K: "Kumano" morphotype, BW: black-white hybrids, BR: black-red hybrids, and RW: red-white hybrids. Putative backcrosses from simulations are represented with three letters, the first two indicate the putative F1 hybrid parent and the third one the putative pure parent. Thus, for example BRB: backcrosses from black-red F1 hybrids and pure black individuals.

and *S. inermis*, one was assigned as a putative hybrid of *S. inermis* and *S. ventricosus*, one was categorized as *S. cheni*, and the last one as *S. inermis*.

The distribution of the maximum Q-scores from the simulated samples indicated that putative pure and F1 hybrids could be clearly discriminated; however, putative backcrosses presented overlapping maximum Q-scores with putative pure and F1 hybrids (Fig. 5). Indeed, putative F1 hybrids presented maximum Q-scores ranging from 0.5 to 0.8, whereas those of putative backcrosses varied from 0.5 to 1 (Fig. 5).

## Genetic divergences between samples with sympatric occurrence

Genetic divergences between species were estimated only using specimens from Hiroshima with clear morphological distinction of each species and non-admixed genetic ancestry in clustering analyses. Using the panel of microsatellite loci, we obtained pairwise FST values

**Table 4 Genetic distances (FST) estimated from D-loop sequences (below diagonal) and 10 microsatellite loci (above diagonal) using individuals with clear morphological and genetic distinction and putative morphological hybrids collected in Hiroshima.**

| | *S. ventricosus* | *S. inermis* | *S. cheni* | Black-white | Red-white |
|---|---|---|---|---|---|
| *S. ventricosus* | | **0.149** | **0.158** | 0.011 | **0.136** |
| *S. inermis* | 0.119 | | **0.118** | **0.127** | **0.138** |
| *S. cheni* | 0.358 | 0.235 | | **0.111** | **0.091** |
| Black-white | 0.003 | **0.104** | 0.254 | | **0.087** |
| Red-white | **0.204** | **0.113** | **0.187** | **0.124** | |

**Note:**
Bold values indicate statistical significance (*P* value < 0.005).

between species ranging from 0.118 (*S. cheni vs. S. inermis*) to 0.158 (*S. cheni vs. S. ventricosus*) (*P* values < 0.001) (Table 4). The D-loop alignment contained 616 bp and was collapsed into 82 haplotypes. All three species and PMH presented haplotype and nucleotide diversities higher than 0.9 and 0.05, respectively (Table S4). D-loop haplotypes were not segregated in separate areas within the haplotype network, nor were they in agreement with the assignment of individuals to their respective origins within a species or putative hybrid group (Fig. 6A). However, all pairwise K2P distances estimated from species and PMH were statistically significant (*P* values < 0.002), except for the comparison between *S. ventricosus* and BW (Table 4). The shortest K2P distance was found between *S. inermis* and *S. ventricosus* (0.119) and the largest between *S. cheni* and *S. ventricosus* (0.358) (Table 4).

The 480-bp alignment of the RH1 gene, including samples from the three species, PMH, and genetically putative hybrids, was collapsed into four haplotypes. All *S. ventricosus* individuals were collapsed into a single haplotype (the main haplotype) highly present in the *S. inermis* (60% of haploid sequences), BW (90%), RW (100%), and "Kumano" (50%) groups (Fig. 6B). Approximately 88% of *S. cheni* individuals had a haplotype differing from the main haplotype by a single mutational step, while 40% of the *S. inermis* individuals carried a different haplotype with one distinctive mutation from the main one. The fourth haplotype was exclusively found in "Kumano" specimens and was derived from the *S. inermis* haplotype. The alignment of RH1 sequences including other *Sebastes* helped us to infer that the three species and the "Kumano" morphotype presented eight common amino acid replacements (*i.e.*, nonsynonymous mutations) at positions 119, 133, 158, 205, 213, 274, 277, and 286 of the rhodopsin protein, with two of them occurring only in this species complex (133 and 286) (Table S5). The mutations exclusively present in *S. cheni* and some "Kumano" individuals were found to cause amino acid replacements at positions 165 (from serine to alanine) and 217 (from methionine to threonine) of the protein sequence, respectively. In contrast, the mutation observed in some *S. inermis* individuals did not alter the amino acid sequence of the rhodopsin protein in relation to the other species of the complex (*i.e.*, synonymous replacement).
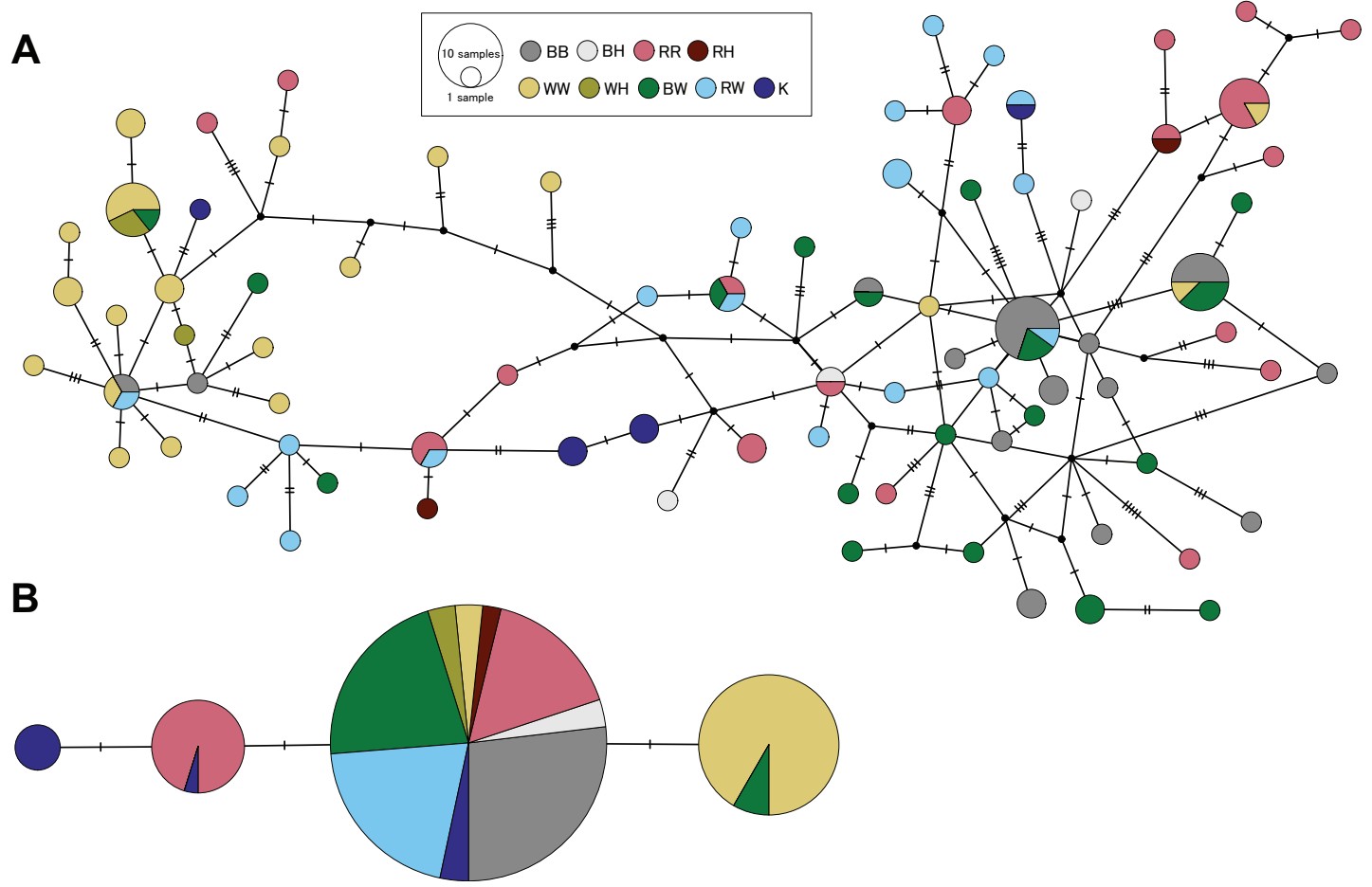

**Figure 6 Haplotype networks constructed from partial sequences of the mitochondrial control region (A) and the intron-free rhodopsin gene (B).** Colours indicate individuals assigned to a single species considering morphological and genetic information. BB, RR, and WW designate individuals identified as *S. ventricosus* (black rockfish), *S. inermis* (red rockfish) and *S. cheni* (white rockfish), respectively. BH, RH, and WH indicate individuals morphologically identified as black, red, and white rockfish, respectively, but genetically classified as putative hybrids. BW and RW indicate specimens classified as putative morphological hybrids of black-white and red-white hybrids, respectively. K is designated to individuals collected in Wakayama Prefecture that display the "Kumano" morphotype.

## DISCUSSION

### Divergences within the species complex

*Kai & Nakabo (2008)* proposed the splitting of *S. inermis* into three species based on differences in colouration, meristic counts, body proportions, and significant genetic divergences estimated from D-loop sequences and AFLP. Our findings support significant genetic divergences in D-loop sequences estimated from sympatric individuals categorized as genetically distinct based on analyses of microsatellite loci. Although we did not use AFLP, the concordant large genetic divergences estimated from D-loop sequences and microsatellite loci highlight the usefulness of the latter as an additional reference for species identification. Moreover, the interspecific differences found here align with those described by *Kai, Nakayama & Nakabo (2002)* in samples from the Seto Inland Sea, Noto (Ishikawa Prefecture), and Wakasa Bay (Kyoto Prefecture). The temporal and geographic

extensions of these interspecific differences highlight the spatio-temporal stability of the species boundaries delimitated using D-loop sequences, despite the likely incomplete lineage sorting or introgressive hybridization suggested by these authors, which also occur in other closely related rockfishes (*Hyde & Vetter, 2007*; *Schwenke, Park & Hauser, 2018*).

The large FST values estimated from microsatellite loci are concordant with divergences between other rockfishes with broad sympatric occurrence (*Roques, Sévigny & Bernatchez, 2001*; *Narum et al., 2004*). The existence of significantly different allele distributions between sympatric species (Fig. S3), and the genetic clusters (in both STRUCTURE and DAPC) concordant with the taxonomic descriptions of the three species suggest that our dataset of microsatellite loci is sufficiently informative to separate them despite the confounding effect of high mutation rates and the multi-step mutation model of these markers, which can possibly lead to congruences in allele sizes (*Morales et al., 2021*).

Among vertebrates, the maximum absorption spectra ($\lambda_{MAX}$) of downwelling sunlight are greatly determined by the type of chromophore bound to the opsin proteins, including RH1, as well as amino acid combinations at specific spectral tuning sites (*Musilova, Salzburger & Cortesi, 2021*). Given the decreasing trend of downwelling sunlight intensity along the water column, nonsynonymous mutations in RH1 suggest that species inhabit environments with different levels of downwelling sunlight owing to divergences in depth distribution (*Musilova, Salzburger & Cortesi, 2021*). Our alignment of the RH1 sequences, which includes other *Sebastes* species, revealed seven amino acid replacements that coincidentally occurred in our focal species and other species inhabiting shallow environments (*Sivasundar & Palumbi, 2010*; *Shum et al., 2014*). For example, a replacement of isoleucine with leucine at position 119 of the RH1 protein has been associated with shifts to shallower environments (*Sivasundar & Palumbi, 2010*), with punctual variation at this position occurring in the "deep" (isoleucine) and "shallow" (valine) groups within the beaked redfish *Sebastes mentella* Travin, 1951 (*Shum et al., 2014*).

The mutations identified in the RH1 gene of our focal species provide insights into their ecological differences in bathymetric distribution, which is consistent with recent ecomorphological analyses (*Deville et al., 2023*). In the case of *S. inermis*, it shares the same amino acid sequence with *S. ventricosus* because the distinctive mutation found in the former does not cause an amino acid replacement in the RH1 protein. Thus, the adaptation of *S. inermis* to deeper environments with lower light intensity is likely manifested through other mechanisms, such as larger relative eye sizes (*Deville et al., 2023*), which enables it to capture more photons (*de Busserolles et al., 2020*). The congruence in amino acid sequences in both species may represent a common adaptation to shallow environments with low light intensity, such as *Zostera L.* and *Sargassum* beds, where *S. inermis* is usually found (*Kai & Nakabo, 2008*) and *S. ventricosus* can occasionally incur (*Shoji et al., 2017*). In contrast, *S. cheni* exhibits a nonsynonymous mutation that leads to an amino acid replacement from serine to alanine at position 165. This nonsynonymous mutation has not been reported in any of the 35 *Sebastes* species with available rhodopsin sequences but has been observed in certain cichlids with $\lambda_{MAX}$ between 498 and 503 nm that inhabit rocky environments in shallow waters of Tanganyika Lake (*Sugawara et al., 2005*). Structural

analysis of the rhodopsin protein has revealed that position 165 is in the 4th transmembrane domain (*Sivasundar & Palumbi, 2010*). Amino acid replacements at this position can alter the dimerization interface of the functional protein, leading to changes in the $\lambda_{MAX}$ (*Schott et al., 2014*; *Ito et al., 2022*). Thus, it is likely that the amino acid replacement at position 165 in *S. cheni* causes changes in its $\lambda_{MAX}$ in response to different downwelling sunlight intensities compared to the other two species. Although the specific $\lambda_{MAX}$ ranges for these species would provide a deeper understanding of their visual adaptations to environments with varying levels of downwelling sunlight, the presence of an amino acid replacement in *S. cheni* underscores the significance of selective pressures that drive ecological diversification within the species complex, adding more evidence to its previous separation into independent species.

## Hybridization within the species complex

Hybridization was inferred from PMH with intermediate morphotypes and population genetic assessments. A total of 29 putative genetic hybrids were detected in our population genetic surveys using clustering analyses with a panel of 10 microsatellite loci. The performance of our STRUCTURE analysis for detecting these putative hybrids relied on a confidence rate of 90%. This is because our number of loci, genetic divergences between the parent species (0.10 < FST < 0.17) (Table 3), and the proportion of putative hybrids in the samples (~8.33%) are close to the ones necessary to attain this rate considering a Q-score threshold value of 0.9 to classify an individual as genetically putative pure or hybrid (*Vähä & Primmer, 2006*; *Sanz et al., 2009*). Based on Q-scores, 14, 13, and two individuals were classified as putative genetic hybrids of *S. cheni* × *S. ventricosus*, *S. cheni* × *S. inermis*, and *S. inermis* × *S. ventricosus*, respectively. It is important to note that the number of putative hybrids inferred from the clustering analyses may be underestimated because we could not include a reference population for each species in the STRUCTURE analysis (*Ravagni, Sanchez-Donoso & Vilà, 2021*). Hence, some genetically distinct individuals with intermediate morphotypes may be backcrosses, as indicated by our simulations (Fig. 5). Considering this and the posterior individual probabilities of DAPC, 50 potential putative hybrids could be inferred from our samples.

The PMH exhibit intermediate colourations and meristic counts but possess otolith weight~age relationships resembling that of *S. cheni*. The presence of intermediate colourations, along with hybridization events, suggest that colouration patterns alone may not be sufficient to maintain reproductive isolation among species (*Gray & McKinnon, 2007*). Other factors, such as specific environmental conditions and assortative mating, may play a role in determining the relevance of colouration patterns for reproductive isolation (*Schumer et al., 2017*; *Pires et al., 2019*).

The network of D-loop haplotypes did not show any clear pattern of discordance with the morphological identification of individuals and results from microsatellite loci, not providing enough evidence to support our expectations of hybridization driven by females. However, the larger genetic divergences between PMH and *S. cheni* might suggest that the former are likely originated from mating pairs wherein a male *S. cheni* mated with a female *S. ventricosus* (BW morphotype) or *S. inermis* (RW and "Kumano" morphotypes). These

mating pairs are consistent with *in situ* observations indicating that females tend to copulate with larger males (*Shinomiya & Ezaki, 1991*), and higher growth rates of *S. cheni* (*Kamimura et al., 2014*). The larger sizes of the PMH can provide a selective advantage during reproductive seasons, as larger males establish larger territories, engage in agonistic behaviour, patrol their territories, and perform courtship when encountering females, in contrast to smaller males (*Shinomiya & Ezaki, 1991*). Hence, the size-assortative mating led by females and selection are important for the persistence of the PMH.

All genetically putative hybrids detected from PMH had the same RH1 haplotype as that of *S. ventricosus*. This observation may indicate introgression of the RH1 haplotypes between species with positive frequency-dependent selection in favour of the haplotype found in *S. ventricosus* (*Sinervo & Calsbeek, 2006*). Considering that hybridization is mediated by females and that the two RW morphotypes slightly differing in colouration (Fig. 2) were only found in two specific sampling sites (Osaki-Shimozima East and Etajima Islands) off Hiroshima, this selection process may be particularly influential in the perception of male colouration by females in dim-light environments such as seagrass beds, in which *S. inermis* and *S. cheni* engage in foraging activities (*Shoji et al., 2017*). In these environments, the persistence of RW individuals is not only explained through assortative mating but also through their higher fitness at foraging and performing defensive responses against predators, as a red-brown colouration may be more difficult to detect within seagrass beds than the red colouration of *S. inermis* (*Deville et al., 2023*).

The behaviour of males during reproductive seasons indicates that they allocate reproductive effort to mating activities rather than to sperm production (*Fujita & Kohda, 1996*). However, the single-brood reproductive strategy (*Plaza, Katayama & Omori, 2004*) and the cases of polygamy reported in the species complex (*Shinomiya & Ezaki, 1991*; *Blanco Gonzalez et al., 2009*) suggest that prezygotic isolating barriers, such as sperm competition, can be important for maintaining the reproductive isolation of species. Noteworthy is the reinforcement of reproductive isolation because of the potential fitness disadvantage of *S. inermis* × *S. ventricosus* hybrids (*Bank, Hermisson & Kirkpatrick, 2012*; *Servedio & Hermisson, 2020*), which are expected to have lower growth rates and reproductive success than the other hybrids described here. Further assessment of assortative mating and hybrid fitness within the species complex could deepen our knowledge of the mechanisms to maintain or reinforce reproductive isolation, especially considering that the release of thousands of juveniles with known ancestry can represent an opportunity to study hybridization *in situ*.

## Speciation-with-gene-flow in the species complex

The patterns of divergence and hybridization observed in these species indicate that they fall into the second and third stages of speciation described by *Wu (2001)*. At these stages, parent species can hybridize to form hybrid swarms (*i.e.*, fertile hybrids with intermediate morphotypes), and their independent evolution in sympatry is maintained through competitive exclusion. In this scenario of speciation-with-gene-flow, introgression does not occur in genomic regions crucial for maintaining species boundaries (*Nosil, Funk & Ortiz-Barrientos, 2009*). The lack of introgression in these regions causes anomalously high

interspecific divergences, such as those observed in the KSs7 and CGN1 loci (FST > 021), leading to the inference that these loci are under putative directional selection (*Nosil, Funk & Ortiz-Barrientos, 2009*). The anomalously high interspecific divergences of these loci, along with low diversity values within each species and deviations from HWE (especially in the CGN1 locus) (Table S2), further support the occurrence of an ecologically selective sweep (*Schlötterer, 2002, 2003; Buonaccorsi et al., 2011*). This type of selective sweep occurs when the variation in a genomic region is reduced or eliminated owing to its proximity to a new beneficial mutation that is increasing in frequency through recent adaptation (*Hermisson & Pennings, 2017*). Another finding suggestive of an ecologically selective sweep is the absence of FST outliers at locus KSs7 in other rockfishes closely related to the *S. inermis* species complex that inhabit the same area (*An et al., 2009*). This is because new advantageous mutations causing adaptive divergence and linked to the KSs7 locus may have appeared more recently. Similar cases of selective sweeps have been observed in closely related rockfishes with different depth distributions (*Buonaccorsi et al., 2011; Behrens et al., 2021; Olivares-Zambrano et al., 2022*) and depth-related ecotypes within a single species (*Saha et al., 2021*). The occurrence of ecologically selective sweeps across *Sebastes* rockfishes indicates that recent adaptation to new environments contributes to the ongoing diversification of species. Therefore, further characterization of the genomic variations surrounding the KSs7 and CGN1 loci is necessary to determine the conditions that promote diversification within the *S. inermis* complex.

## The "Kumano" morphotype

The combination of morphological features of "Kumano" explains why local fishermen consider this morphotype as a "big variant" of the red-coloured rockfish *S. inermis*. Although genetic divergences were not estimated due to the low number of individuals, the D-loop haplotypes indicate that the "Kumano" specimens are part of the species complex. Analysis of the Sebi1 locus, used by *Deville et al. (2023)* to discriminate *S. ventricosus*, suggested that this morphotype does not possess the typical alleles of *S. ventricosus* (>160 bp) (Fig. S3). In terms of the two loci under putative divergent selection, the "Kumano" specimens carried the most frequent allele of *S. inermis* at the CGN1 locus and some exclusive alleles at the KSs7 locus (Fig. S3). STRUCTURE analysis suggested a possible hybrid origin for the "Kumano" specimens, with approximately 75–82% of their ancestry corresponding to *S. inermis*, along with 10–16% ancestry of *S. cheni* in three individuals, and 16% of *S. ventricosus* in one specimen. However, when four genetic clusters were inferred using STRUCTURE analysis, "Kumano" specimens and the RW morphotype were grouped together in a separate category with high Q-scores (Fig. 3). Additionally, a punctual amino acid replacement was observed at position 217 in some individuals, resulting in an amino acid replacement from threonine to methionine regarding the three species of the complex (Table S5). In other rockfishes, this amino acid replacement has been associated with shifts in shallower waters (*Sivasundar & Palumbi, 2010*). The position 217 falls under the 5th transmembrane domain, and possible changes in this position are related to modifications in the $\lambda_{MAX}$, which is related to visual sensitivity (*Schott et al., 2014*). These findings suggest that the endemic "Kumano" morphotype might have

exclusive alleles at loci responsible for maintaining species divergence in the presence of gene flow within the species complex. Considering this evidence, the hypothetical hybrid origin of the "Kumano" morphotype aligns with theoretical models predicting that hybridization, combined with intermediate assortative mating and low variation in reproductive success, could act as a potential mechanism for rapid evolution in specific environments (*Baskett & Gomulkiewicz, 2011*). The first condition, intermediate assortative mating, is fulfilled in this species complex, whereas the second depends on the level of preference of females for the "Kumano" morphotype, which is considered "rare". A comprehensive morphological and genetic characterization of more individuals is necessary to deeply assess this hypothetical hybrid origin and support the emergence of "Kumano" as an incipient species resulting from the ongoing process of speciation-with-gene-flow within the *S. inermis* complex.

## CONCLUSIONS

The dynamics of divergence and hybridization within the *Sebastes inermis* complex (*Sebastes cheni*, viz. *Sebastes inermis*, *Sebastes ventricosus*, and their putative morphological hybrids (PMH)) was assessed using sequences of the mitochondrial control region (D-loop), the intron-free rhodopsin (RH1) gene, and 10 microsatellite loci. We anticipated that each species would maintain its genetic divergence even in the presence of hybridization, and that the PMH would exhibit genetic admixed ancestry of the parent species. We found large genetic divergences in D-loop, distinctive mutations in the RH1 gene, and three genetic clusters obtained from microsatellite loci, which are concordant with the morphological description of each species. These findings can be used as additional information to facilitate species identification. *S. cheni* is the only species with a nonsynonymous mutation in the RH1 gene, which suggests differential adaptations of this species to dim-light environments. Two microsatellite loci under putative divergent selection suggest that they are possibly linked to genomic regions, wherein interspecific gene flow is typically restricted because they are crucial for maintaining species boundaries. Further characterization of the genomic regions surrounding these loci is underway. A total of 29 putative genetic hybrids were detected using microsatellite loci. The higher growth rates of the PMH, genetic divergences between species and the PMH using D-loop, and mating behaviour of females and males during reproductive seasons suggest that the PMH can present higher fitness than the parent species in certain environments.

The genetic admixed ancestry and nonsynonymous mutation in the RH1 gene of the PMH known as "Kumano", provide support for the potential contribution of hybridization in generating novelties within the *Sebastes inermis* complex. Overall, this study indicates that the genetic divergence of each species within the complex is maintained in the presence of hybridization.

## ACKNOWLEDGEMENTS

We would like to thank all lab members, Gou Uehara and Shunji Uehara, who kindly provided samples and helped with logistical support to ease the collection of specimens.

Our gratitude is extended to Hirosuke Kimura, Naoyuki Nakase, and Keisuke Doi for providing the "Kumano" samples.

### Funding

Diego Deville was supported by the Japanese Government through the Ministry of Education, Culture, Sport, Science, and Technology (MEXT) fellowship. The funders had no role in study design, data collection and analysis, decision to publish, or preparation of the manuscript.

### Grant Disclosures

The following grant information was disclosed by the authors:
Ministry of Education, Culture, Sport, Science, and Technology (MEXT).

### Competing Interests

The authors declare that they have no competing interests.

### Author Contributions

- Diego Deville conceived and designed the experiments, performed the experiments, analyzed the data, prepared figures and/or tables, authored or reviewed drafts of the article, and approved the final draft.
- Kentaro Kawai conceived and designed the experiments, performed the experiments, authored or reviewed drafts of the article, and approved the final draft.
- Hiroki Fujita conceived and designed the experiments, authored or reviewed drafts of the article, and approved the final draft.
- Tetsuya Umino conceived and designed the experiments, authored or reviewed drafts of the article, and approved the final draft.

### DNA Deposition

The following information was supplied regarding the deposition of DNA sequences:

DNA sequences of the mitochondrial control region and the intron-free rhodopsin gene are available at GenBank: OR209356 to OR209485, and OR209492 to OR209621, respectively.

The sequences are also available in the Supplemental Files. The amino acid alignment of the RH1 protein including Sebastes species and the "Kumano" morphotype is also available in the Supplemental File.

### Data Availability

The datasets of microsatellite loci, and alignment of DNA and amino acid sequences are available at figshare: Deville, Diego (2023). Genetic divergences and hybridization within the *Sebastes inermis* complex. figshare. Dataset. https://doi.org/10.6084/m9.figshare.23294375.v2.

## Supplemental Information

Supplemental information for this article can be found online at http://dx.doi.org/10.7717/peerj.16391#supplemental-information.

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
