# Peer review of "Genetic divergences and hybridization within the Sebastes inermis complex"

_PeerJ, doi:10.7717/peerj.16391_

## Round 0.1 · original submission · Major Revisions

Authors have written a comprehensive genetic and morphological study on the Sebastes inermis species complex. Your results suggested a model of speciation-with-gene-flow, where hybridisation serves as a mode for gene exchange. Reviewers have given their comments and suggestions, authors please follow their reviews and carefully answer all their questions. English needs to be improved, all reviewers suggested this point. Particularly, Reviewer #4 strongly criticized the manuscript, however, authors need to defend the questions and provide your appropriate response.

**Language Note:** The Academic Editor has identified that the English language must be improved. PeerJ can provide language editing services - please contact us at copyediting@peerj.com for pricing (be sure to provide your manuscript number and title). Alternatively, you should make your own arrangements to improve the language quality and provide details in your response letter. – PeerJ Staff

Reviewer 1 ·

Basic reporting

Overall manuscript is well explained and English used throughout the manuscript is good. Context and the article structure is well prepare. Tables and figures were adequate to explain the findings and to discuss the results. However, the references used in this manuscript quite old and many of them is more than 5 years. Some minor revisions need in order to change the in text citation with the new references.

Experimental design

This research manuscript is well design and align with the aims/ scope of this journal. The research gap in the sebastes complex can be fills from this manuscript. Authors can explained a little bit about the sampling method or collection. eg: Fishes were collected from which bank? No DNA concentration were explained in this manuscript.

Validity of the findings

Findings were good and can be replicate. Robust statistical analysis has been done and explained throughout the manuscript. It is well written and easy to understand.

Additional comments

I am not happy with the references. Many of them is outdated, in which more than 10 years and only few of it is less than 5 years. Please double check and find new reference to support the findings of this research.

Annotated reviews are not available for download in order to protect the identity of reviewers who chose to remain anonymous.

·

Basic reporting

No comment

Experimental design

No comment

Validity of the findings

No comment

Additional comments

Deville and colleagues deliver a comprehensive genetic and morphological investigation of the Sebastes inermis species complex, which includes three Sebastes species: S. inermis, S. cheni, and S. ventricosus. Through the sequencing of the mitochondrial control region, the rhodopsin gene (associated with dim light vision), and 10 microsatellite loci, they unravel intriguing patterns of speciation, hybridisation, and introgression. The results suggest a model of speciation-with-gene-flow, where hybridisation serves as a mode for gene exchange. Particularly noteworthy are two loci, KSs7 and CGN1, identified as being under divergent selection, suggesting their possible linkage to crucial genomic regions that define species boundaries and restrict gene flow.

The authors also shed light on the potential emergence of a new species within the complex, referred to as the "Kumano" morphotype. This morphotype exhibits a composite genetic makeup, with roughly 75-82% of its genetic ancestry tied to S. inermis and S. cheni with traces of S. ventricosus, hinting at a hybrid origin. The authors underscore the practical implications of their findings, hinting at the need to assess the potential impact of stock enhancement programs, which might inadvertently augment the chances of introgression within the species complex.

I applaud the authors for the insightful exploration of the speciation and introgression dynamics within the Sebastes inermis complex, offering yet another intriguing case of hybrid origin in Sebastes species. The manuscript is well written and the analyses carried out are solid and robust.

minor comments:
Line 206: "seg" should be replaced with "sec".

Line 380: Avoid the term "genetically pure". The significant introgression and moderate Fst values reported in this study suggest a more intricate picture. Consider using "genetically distinct" instead.

Line 358: Could you clarify if any RH1 chromatograms indicated heterozygosity via overlapping peaks?

·

Basic reporting

1. The language is lucid and easy to comprehend, with the exception of some terms that have been highlighted in the annotated pdf (attached)

2. Additional reference on the D-loop region in order to justify the selection of this region for the detection of hybridization events.

3. The material and methods section MUST contain a statement "all sequence data was deposited at the NCBI GenBank and assigned accession numbers".

Hypothesis has been stated.

Experimental design

The authors have applied three approaches to come to a conclusion regarding introgression.
1. Microsatellite DNA
2. D-loop (Mitochondrial DNA hypervariable region)
3. Rhodopsin gene.

These are my comments regarding the above:
1. In the case of suspected cryptic hybrids, the mtDNA barcoding sequences can be used to identify individuals with unique DNA sequences.
2. The D-loop is hypervariable and it may not be possible to assign a particular sequence to a sub-species as there can be significant variation within individuals of the same species.
3. The term F1 hybrid cannot be used as the specimens have been obtained from the wild and they may be from any generation. How have the authors come to the conclusion that they are F1? the correct term should be putative F1 as evidenced from the statistical analysis.

Validity of the findings

The article is well written, but needs an element of synthesis as the data is obtained from three different sets of loci.
The authors should state if the conclusions from each data set corroborate each other or are divergent.

Additional comments

The article is well written and reports on an extensive study using multiple DNA loci.

·

Basic reporting

The language and use of English needed much improvement. Here are some examples of unclear language:
“..which present extensive sympatry”
“Introgression –free rhodopsin gene”
“intro-free rhodopsin gene” (Intron – free)
The Introduction did not sufficiently support the objectives or give enough background to understand the project. More literature citation and more recent articles are needed.
For example” Where the three species are stocked is not clear – The authors discuss areas where introgression is most likely – do these areas overlap?
The background explanations and references on using genetic markers for species identification were unclear. Why these methods were not employed or how were these methods an improvement? For example: Two microsatellite markers and dissimilar allele patterns of AFLP but those markers were not employed in this paper.

That paper, self-contained, did not have relevant results to hypotheses.

Experimental design

The introduction discusses reproductive isolation of genes involved in speciation but this study was not designed to detect genes involved in speciation. One gene (RH) was pulled from the literature to sequence. More genetic data, such as whole genomic sequencing, would be a better method to explore island of selection in the genome. Authors did not investigate which genes could be under selection but did discuss outlier loci and divergent selection and genome-wide patterns of divergence, but with just 10 microsatellites) that is not applicable but that is not a genome wide scan of areas of divergence,
It was a major flaw for the authors to verify ‘pure’ reference species by using the genetic clustering analysis. This is a problem because the genetics should be independent data to morphology data when making inferences on introgression. Choosing morphological types in areas of non-sympatry would be a better way to determine reference species. Using genetics in the analysis and in the methods is circular and not valid.
These three species are obviously closely related, recently diverged and cannot be resolved by D-loop (Control Region). Using polymorphisms in the control region to investigate introgression is problematic because of shared ancestral polymorphisms retained in sequence data.
The discussion: “ our findings support significant genetic difference in D-loop sequences of individuals occurring in sympatry” .The data files for D-Loop only have morphology designation and no location data so there was no way to evaluate result or conclusion.
In addition, using microsatellite genotypes cannot differentiate between incomplete lineage sorting and introgression. Using linked SNPs or sequence data is a more robust way to differentiate introgression from incomplete lineage sorting.
Objective 3 and 4 are a big reach beyond the data and methods used for this paper.

Validity of the findings

The sequence data files did not included collection locations yet the location of collections were important to the conclusions.
The microsatellite data did not conform to the neutral assumptions required to make conclusions for structure (cluster) analysis nor for the coalescent analysis.
The likelihood of microsatellites ascertainment bias was not evaluated. How efficient were these markers across species? Could null alleles bias these results? Most markers were developed in other Sebastes species then the once for this study.
The authors reference another Sebastes introgression paper that used microsatellite (Buonaccorsi et al 2005), but that study evaluated 65 loci.
Again, it is problematic to include microsatellite loci in clustering analysis with significant signals of divergent selection. Also this analysis used loci that were in HW disequilibrium and one association of linkage which will bias results of cluster. Here is another bias to using cluster data to ‘assign’ species groups.

The conclusions were not supported by the data.

---

## Round 0.2 · Minor Revisions

The manuscript has been advised for some minor revision by the reviewers, please follow the reviewers comments carefully before your submission. Reviewer#3 has advised the authors to include the pictures of representative specimens, hence authors please follow up that suggestion in addition to improve the English language of the article.

Reviewer 1 ·

Basic reporting

Overall, the authors has made revision according to the suggestion and answering all questions in the previous review process. English is ok and the structure of the article is appropriate and relevant.

Experimental design

Experimental design was well explained and revise according to the suggestions.

Validity of the findings

All data is well explained throughout the article and supported by the recent references.

Additional comments

Overall is ok, authors has made changes according to the suggestions and explained clearly.

·

Basic reporting

1. The language can be comprehended by a peer who has experience in the field of population genetics.
2. The literature review is sufficient and has been improved upon from the first draft.
3. The article is structured in a style that makes it readable. There are no photographs of the specimens.
4. Self contained and findings can be related to hypothesis.

Experimental design

1. The experimental design is in concordance with population genetic studies.
2. Both nuclear DNA (Microsatellites) and Mitochondrial DNA (Control regions) have been investigated.
3. The investigation is detailed and rigorous.
4. Methods can be replicated at a different geographic location with different specimens.

Validity of the findings

1. Meaningful article.
2. Yes, all data provided as fasta files and deposited at public databases. Release data upon publication of this manuscript.
3. Conclusions section synthesizes the findings and aligns them with the hypothesis.

Additional comments

There are no images of the representative specimens.
Please include these images as a reader will be interested in the different morphotypes.
Use an appropriate scale.

---

## Round 0.3 · accepted · Accept

The manuscript has been improved well by following the reviewers' comments. Hence, it can be accepted for publication. Authors can express their appreciation to the reviewers in their acknowledgments.